# Experimental and Numerical Study of Orifice Coefficient of Cargo Tank Design of LNG Vessels

**Se-Yun Hwang [1], Kwang-Sik Kim [2], Ho-Sang Jang [2] and Jang-Hyun Lee [2,*]**

[1] Research Institute of Industrial Technology, INHA University, 100 Inha-ro, Michuhol-gu, Incheon 22212, Korea; seyun.hwang@gmail.com

[2] Department of Naval Architecture and Ocean Engineering, INHA University, 100 Inha-ro, Michuhol-gu, Incheon 22212, Korea; Kskim00@gmail.com (K.-S.K.); wa5211@hanmail.net (H.-S.J.)

\* Correspondence: jh_lee@inha.ac.kr; Tel.: +82-32-860-7345

**Abstract:** Liquid cargo storage tanks of liquefied natural gas (LNG) carriers are designed by strict standards to maintain the cryogenic state ($-163\,^\circ$C). For most LNG cargo storage tanks, it is mandatory to install a system that can safely store leaked fluid for 15 days in the case of leakage of liquid cargo due to crack of the insulation system. To ensure safety, it is necessary to predict the amount of LNG spilling from the cracks in the insulation panels. Although international regulations are provided, they rely on a conservative and consistent coefficient. In this study, experimental and numerical methods were applied to examine the design factor used to predict the flow rate in the tank design process. To check the amount of leakage that occurs under pressure conditions of LNG tanks, an experiment was conducted using crack specimens and pressure containers filled with water. In order to simulate the leakage of LNG, the amount of leakage was predicted using the Computational Fluid Dynamics (CFD) method. The distribution of leakage quantity was investigated according to the shape of the crack through the pressure vessel experiment and the analysis. Through CFD analysis, the leakage rate of LNG was calculated for each operating pressure condition through the crack. Finally, the results of this study examined the need to identify and reconsider the coefficients due to international guidelines and other factors in calculating orifice coefficients applied to the design of LNG tanks.

**Keywords:** orifice coefficient; LNG leakage; LNG cargo containment system; leakage test; computational fluid dynamics

## 1. Introduction

The designing liquefied natural gas (LNG) tanks to be installed in LNG vessels requires a high level of design and safety evaluation to transport the cryogenic fluid cargo. The type of tank system is representative of a membrane type prismatic tank (type A) and independent tank (type B) [1]. In the case of the independent tanks, since they have structures that are vulnerable to a fatigue damage, they are designed in consideration of high safety for cracks. There have been many studies on the membrane cargo containment systems (CCS) due to widespread application [1,2]. However, the independent CCS has not been used primarily in the past, and demand has been increasing in recent years. Therefore, there is little research related to the design technology of the independent CCS. The independent LNG tanks (type B) are designed and manufactured through precise design verification stages, such as wave load calculation, detailed stress analysis, fatigue analysis, and thermal stress analysis [3]. In addition, there is a possibility of fatal damage if the LNG leakage is during operation time of the LNG vessels. Therefore, design techniques are needed to prevent cracking and catastrophic damage in the initial design stage [4]. The LNG CCS sets up primary and secondary barriers for storing liquid cargo in order to ensure safety cargo transportation. This shall

be designated as a regulation by the international maritime organization (IMO), an international regulation [1]. Furthermore, a second barrier with a drip tray system is installed to store discharged liquid cargo for a predetermined period and ensure the safety of the tank for a period of operating time [3,4]. This design method is generally known as the leak before failure (LBF) concept. In the event of a crack in the tank as shown in Figure 1, the leakage of LNG cargo from the containment system is safely stored in the drip tray even if it does not spread rapidly within the specified days (15 days) [1]. This is designated as an international regulation that must be applied when designing an LNG cargo containment system (CCS) [1].

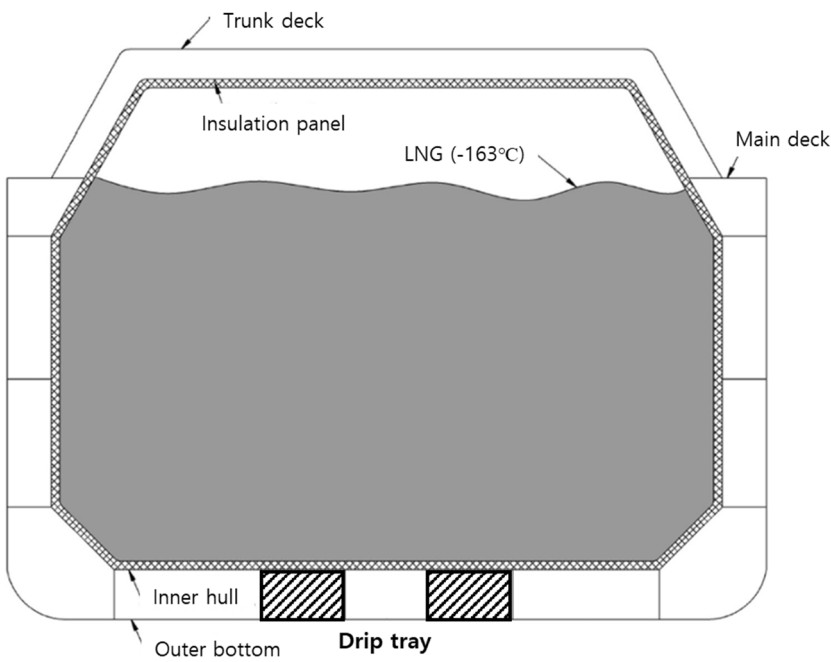

**Figure 1.** Configuration of independent liquefied natural gas (LNG) tank with drip tray.

During the design of the containment system, the leakage rate (primary barrier) through cracks in the external shell plate shall be generally determined for the size of cracks in the cargo tank. However, the international regulations also state that the CCS design process is generally determined by the orifice formula presented in hydrodynamics. In particular, the orifice coefficient uses 0.1 as the experimental result presented by the design guidance [1]. This is the main factor in the drip tray capacity design for safe storage of discharged fluid cargo. Therefore, in the drip tray design stage, the leakage rate for representative crack size should be calculated and its capacity calculated to determine the volume of the drip tray. The assessment of liquid cargo leakage in the tank system represents an important aspect, which is determined by the orifice coefficient [1].

In the conservative view of the general design process, taking into account the greater leakage of LNG liquid is a general approach to dimensioning the size of the leak protection system. The general design process for the LNG tanks presented by the design guidance and shipyard engineer is based on a simple theoretical and empirical method of calculating the orifice coefficient of liquid cargo based on hydrodynamic theory [5–10]. The equation presented in this guidance is the same theory as the equation presented in consideration of the flow of fluid mechanics and the geometric characteristics of orifice shapes [1,5]. However, in the case of the LNG tank loaded with the fluid cargo, the crack size and operating conditions vary for the conditions under which the leakage occurs, so it will be necessary to apply several approaches with an experiment or numerical analysis method considering the leakage characteristics of the fluid for various operating conditions.

The basic concept of this study is to examine in an experimental and numerical method whether the design formula applied at the design stage is reasonable if cracks occur in LNG tanks. It is an

experimental review, especially of the parameters used in the design equation, orifice coefficient. Even in hydrodynamics, orifice flow is based on simplified flow and fluid properties. Since it is difficult to consider the effects of complex multidimensional viscosity of fluid flow, it is generally reflected in empirical form. This theory is also applied in the design of the drip tray of LNG tanks.

In the study related to the orifice of fluid, the fluid leakage is generally found in a variety of studies of physical and apparent losses associated with accidents. These orifice cases have a significant impact on the economic and operational management of the relevant systems. Therefore, because there are significant failures and manifestations, various research cases can be found (e.g., actual fluid burst) [10–15]. Even in these studies, for estimating the liquid losses deriving from the events, hydraulic characterization of the leakage is required, such as the relation between the leak outflow and the hydraulic head, the geometric features of the hole, and the mechanical characteristics of the tank. In addition, for compressible fluids, an empirical expansion factor is applied to the discharge coefficient equation to adjust for the fluid density variation due to changes in pressure upstream and downstream of the orifice plate. Thus, the orifice coefficient should be dependent on the different conditions [5].

This study examined the orifice rate calculation method proposed by the Det Norske Veritas (DNV) class guidance as an international guidebook for the estimation of the orifice rate in the design process of the LNG tank. However, it is impossible to measure the amount of liquid cargo leakage in an LNG tank where the actual crack occurred. Therefore, experimental equipment was installed in the similar condition of the operating conditions of the LNG vessels, and the orifice flow weight was measured using crack specimens. In addition, the crack specimens applied to the flow-rate measurement have too many cases to consider the actual crack geometry and have experimental limitations in processing them. Therefore, the size of the crack was assumed and applied in a rectangular shape based on the typical crack area applied in the LNG CCS strength assessment. In order to simulate the loading condition of the LNG tanks, pressure containers were manufactured, and the internal pressure conditions of the container were adopted to the design pressures of LNG vessels. In the experiment, the internal pressure of the pressure container was adjusted, and the liquid effects were reflected by using water. In order to reflect the effects of LNG, a series of numerical analyses were performed through CFD (Computing Fluid Dynamics) simulations to approach reasonable results [10,11,16,17]. In order to investigate the validity of CFD model, the leakage flow weight and analysis results were compared through experiments using water first, and the CFD model assuming the incompressible fluid was confirmed to be reasonable. Therefore, simulations applied with the fluid characteristics of LNG were carried out using the proven CFD model. Then, the results were analyzed to confirm orifice characteristics of LNG in the tank.

The purpose of this study is to provide the basis for the review of the orifice coefficient applied in the design of the LNG tank. Therefore, the flow rate of liquid leakage was simulated when cracks occurred in LNG tanks through experiments and CFD analysis. To simulate several leakage states, an incomprehensible CFD model was applied, which assumes that the mass flow rate through an orifice is based on mass conservation [17,18]. Of course, although the experimental conditions and numerical analysis methods reviewed in this study differ from those of the actual LNG tank, they were considered to reflect the operating conditions and the characteristics of fluids flowing through the cracks in LNG vessels. As a result, it was confirmed that the size of the crack area was small, the amount of effluent from the fluid was nonlinear, and the leakage was somewhat greater than the values presented in the design guidance. In this study, it is hoped that these results will be reviewed and used as a reference for the safe cargo tank design of LNG vessels.

## 2. Orifice Coefficient

Figure 2 describes the concept of orifice plate geometry and orifice flow as described in hydrodynamics. Assuming that the fluid flow in the pipe is an incompressible fluid, the flow behind the cracks causes the fluid velocity and flow rate to change dramatically due to the crack. As a result, the pressure distribution in the pipe is also decreased from P1 to P2. The Bernoulli equation is

derived based on the energy conservation law in fluid mechanics, and the equation for calculating the leakage rate through the crack is defined as Equation (1). In this equation, the amount of LNG flowing through the cracks is influenced by the orifice coefficient ($C_{orifice}$), which is assumed to be the leakage flow of the liquid cargo:

$$Q_{Leak} = C_{orifice}A \sqrt{2g\left(h + \frac{p_1 + p_2}{\gamma}\right)} \tag{1}$$

where $Q_{Leak}$ is the leakage flow rate(liter/sec); $A$ is the cross-sectional area of the crack; $h$ is the pressure head of the fluid in the tank at the crack position; $\gamma$ is the specific gravity of the discharged liquid; and $p_1$ and $p_2$ are the internal and external pressures of the tank, respectively [1,5].

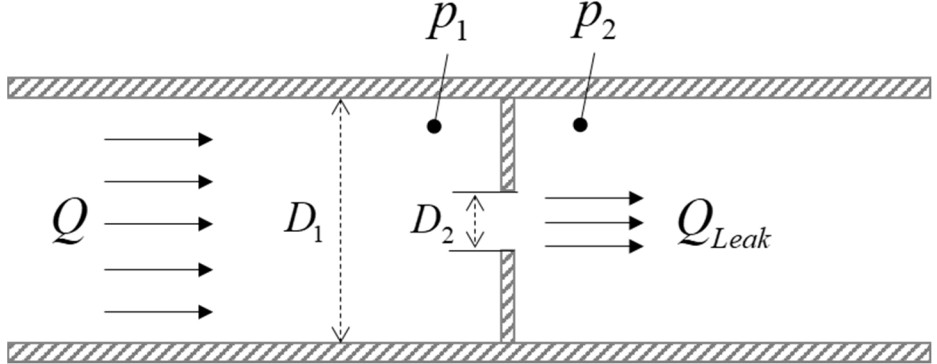

**Figure 2.** Configurations of orifice flow.

This equation strictly holds for an incompressible steady-state flow. It describes the flow with good accuracy for laminar and turbulent flow conditions. The actual form of the fluid flow strongly depends on the geometry of the restriction (particularly for whether it is sharp-edged), and small disturbances may lead to a change from laminar to turbulent flow conditions. However, assuming that this local flow variation is negligible in this study, the equation was utilized even when the flow might have been the orifice flow, as shown in Figure 2. The experiment and the CFD analysis in this study were also assumed to be fluid flows as an incompressible fluid.

## 3. Leakage Experiment

### 3.1. Experiment Setup

An experiment was conducted using pressure vessels and crack specimens to quantitatively measure and examine the leakage flow rate through a crack. Thus, the general size of the crack area considered in the design of LNG tanks was adopted and applied to make the experimental results as general as possible. For the ease of experimentation, cracks were processed and manufactured in a rectangular form. To measure the leakage flow through cracks in a liquid cargo tank, it is important to first study the analogy between an orifice flow and leak flow [1,5]. Therefore, a cracked specimen with a rectangular shape was installed as a typical crack area, and a leakage experiment was performed. Figure 3 shows the configuration of the experimental equipment. The shape of the cracked specimen was rectangular. The crack specimen had a thickness of 10 mm and a diameter of 100 mm. The types of specimens used in the leakage experiment are shown in Table 1.

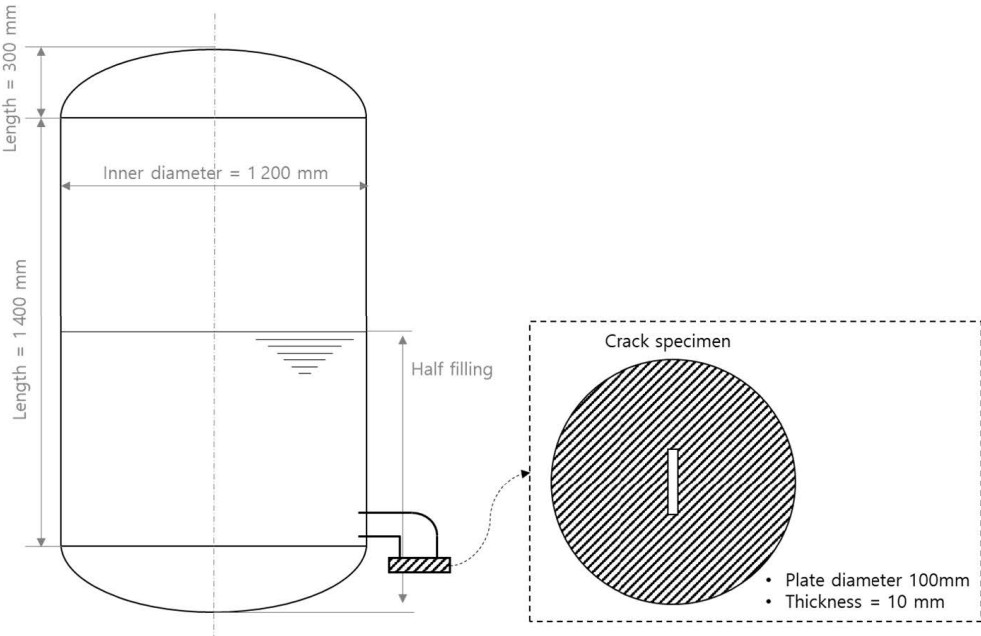

**Figure 3.** Configuration of pressure vessel with crack specimen.

**Table 1.** Type and dimensions of crack specimens.

| No. | Crack Shape | Dimensions [mm] | Area [mm$^2$] |
|:---:|:---:|:---:|:---:|
| 1 | Rectangular | $0.5 \times 7$ | 3.5 |
| 2 | Rectangular | $1.0 \times 7$ | 7 |
| 3 | Rectangular | $1.0 \times 14$ | 14 |

The real-time leak experiment is based on the real operating conditions of an independent LNG tank. The experiment was designed to measure the pressure head and pressure of the fluid inside the tank. To calculate the flow rate, the weight of the fluid that spilled out was measured in real time. The internal fluid is water at room temperature. The pressure inside the pressure tank was configured to maintain each loading condition, and three pressure sensors were installed to measure the pressure inside the tank at the location of the crack specimen. Figure 4 shows a configuration of the experimental equipment. The compressor takes air from a storage tank, pumps it through a pipe, and maintains steady state. Pressure sensors and transmitters were used just upstream of the leak to measure the pressure head at the leak.

Pressure and weight sensors were installed to measure the internal pressure of the tank as shown in Figure 4. Analog and digital pressure sensors were installed at each position. The first pressure sensor (P-Sensor 1) was installed in the upper part of the tank to measure the pressure of the gas (air) inside the tank under half-filled conditions. The second sensor position (P-Sensor 2) was set up to measure the pressure of the fluid in the tank. Finally, the third sensor position (P-Sensor 3) was installed on the crack specimen to measure the orifice flow rate. The flow rate was measured using the weight sensor (W-Sensor) of the fluid flowing out.

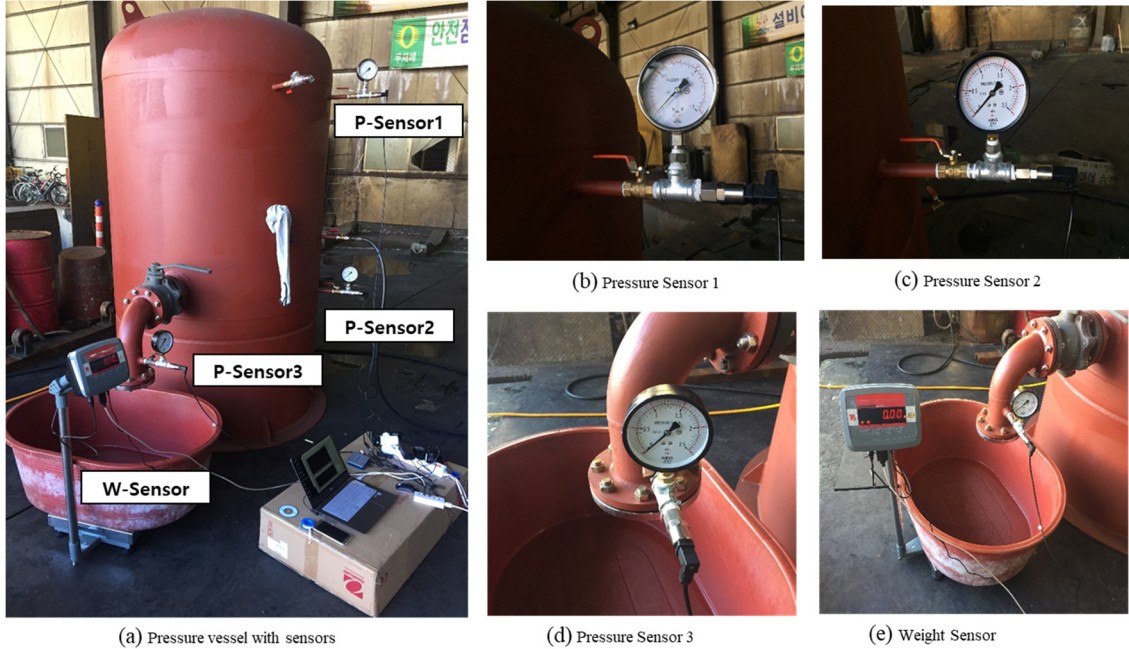

**Figure 4.** Experimental setting of pressure vessel with crack specimen and sensors.

*3.2. Load Cases*

The load conditions for the experiment were selected by considering the load cases (LC) applied to the design of the LNG tank. Generally, a vapor pressure of 0.7 bar is generated when LNG at −163 °C is loaded. Therefore, the first load condition was set at 0.7 bar. When the liquid cargo is loaded in the LNG tank, a depth of more than 25 m (general pressure height loaded with fluid cargo in LNG vessels) is loaded, and static fluid pressure occurs due to the liquid cargo. Therefore, the load condition was selected under load conditions of 2 bar [1]. Finally, 3 bar was used based on the acceleration condition of the LNG tank due to hull motion during operation of the LNG vessel. Table 2 shows the loading conditions applied in the experiment.

**Table 2.** Load cases for leakage experiment.

| Load Case | Pressure | Description |
|:---:|:---:|:---:|
| LC 1 | 0.7 bar | Vapor pressure |
| LC 2 | 2 bar | Hydro static pressure (LNG 25 m) + Vapor pressure |
| LC 3 | 3 bar | LC 2 + Ship acceleration (Assumption) |

The following loading conditions were set for the initial pressure conditions, and the outflow was measured for each loading condition for up to one hour. As the fluid flowed through the crack specimen, the pressure inside the tank gradually decreased. However, since it is difficult to control the amount of pressure change during the experiment, it was assumed to be negligible and was ignored.

*3.3. Experiment Results*

The flow rate was measured for each crack area and the load condition. In the case of specimen 1 (Table 1), 96.3 kg/ of total fluid weight leached out during the 1-hour period under LC1, and the flow rate increased with increasing pressure. Figure 5 shows the results of specimen 1 under LC1. The results of LC2 and LC3 of specimen 1 are shown in Figures 6 and 7. During the experiment, the weight of the fluid leaking out through the cracks was linearly distributed over all the load conditions. As mentioned, the pressure distribution gradually decreased with increasing flow rate, but the pressure changes were ignored.

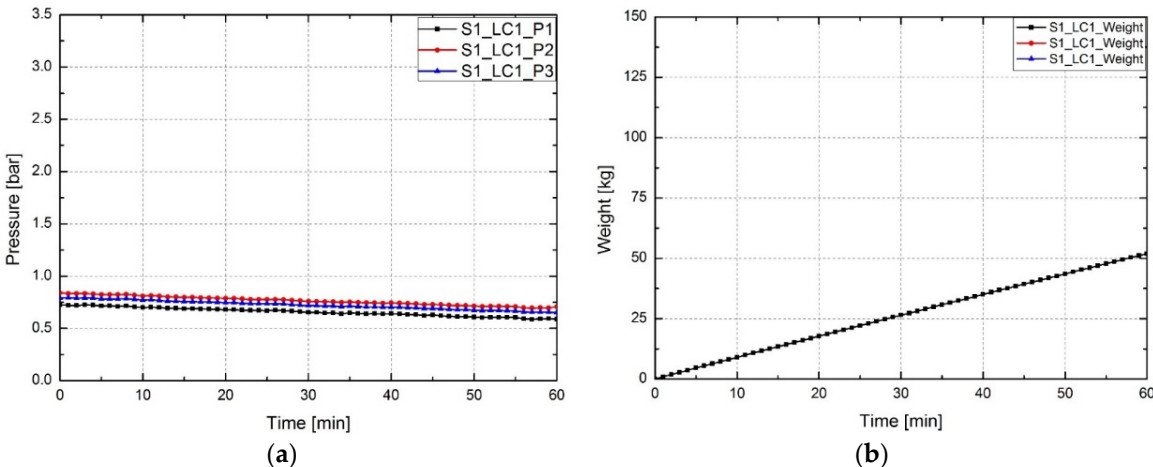

**Figure 5.** Experiment results of $0.5 \times 7$ mm$^2$ specimen for load case 1. (**a**) Pressure histories. (**b**) Fluid weight histories.

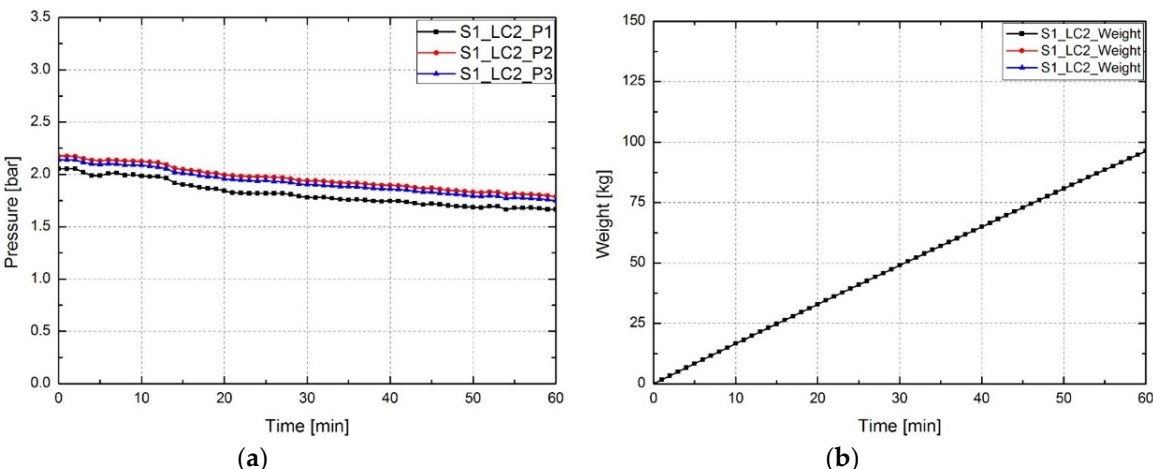

**Figure 6.** Experiment results of $0.5 \times 7$ mm$^2$ specimen for load case 2. (**a**) Pressure histories. (**b**) Fluid weight histories.

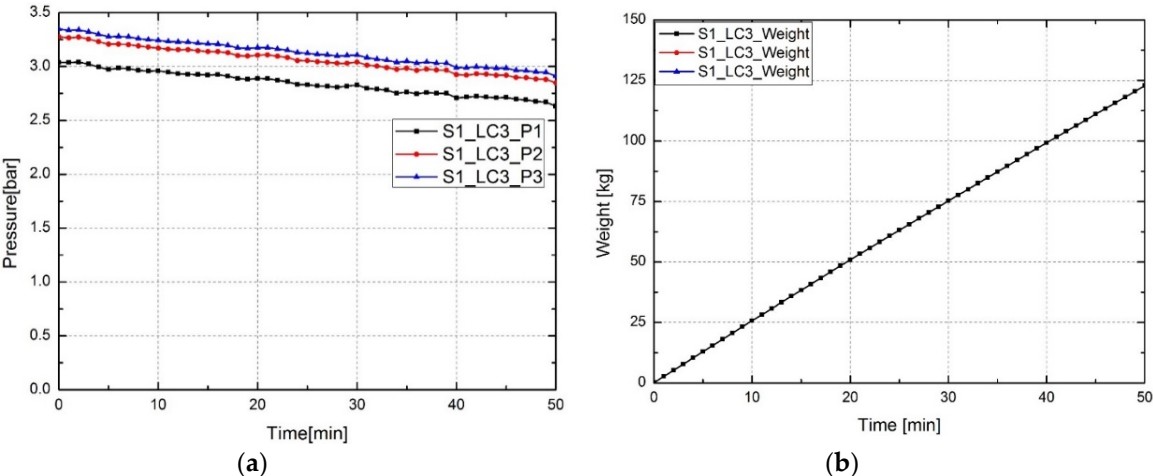

**Figure 7.** Experiment results of $0.5 \times 7$ mm$^2$ specimen for load case 3. (**a**) Pressure histories. (**b**) Fluid weight histories.

The flow weight per minute was calculated since the end time was different for each experimental condition. Table 3 shows the fluid weights per minute for each experimental condition. Figure 8 shows

the variation of the flow rate per minute according to the pressure condition for each crack specimen. Considering the experimental error, the pressure condition and the leakage flow rate were found to be proportional to the size of the specimen area.

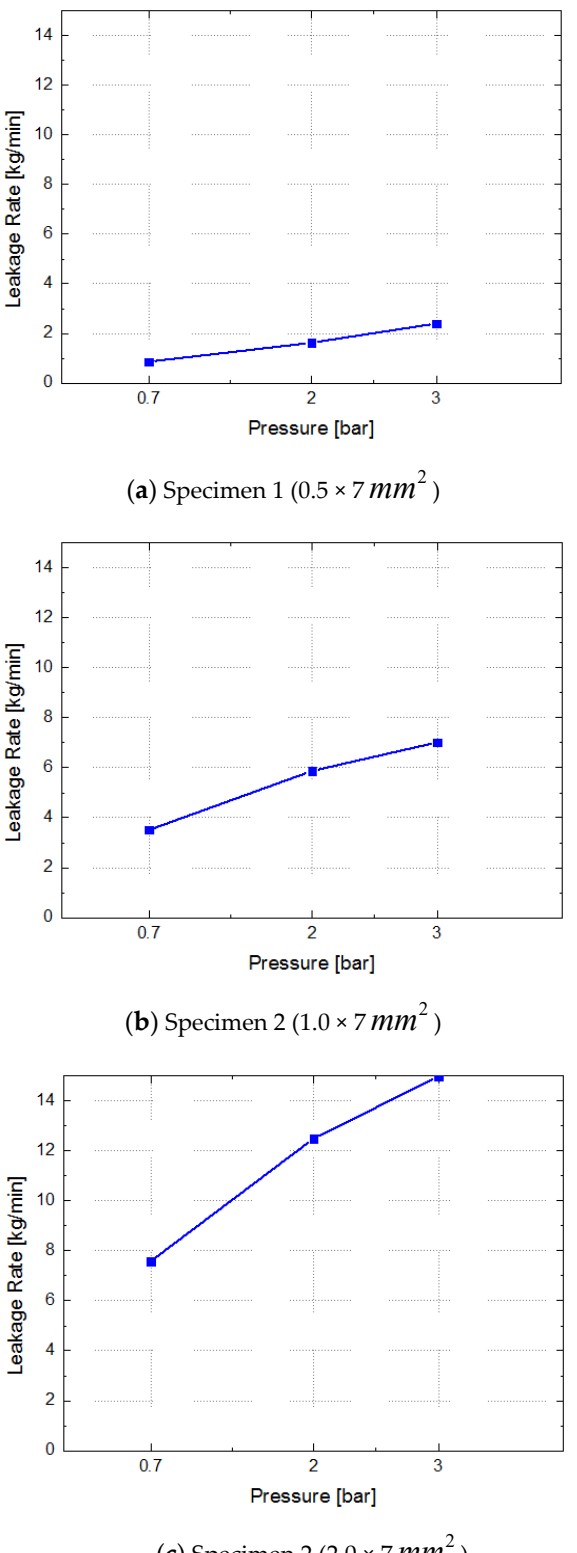

(**a**) Specimen 1 (0.5 × 7 $mm^2$)

(**b**) Specimen 2 (1.0 × 7 $mm^2$)

(**c**) Specimen 2 (2.0 × 7 $mm^2$)

**Figure 8.** Leakage flow rate according to pressure conditions. (**a**) Specimen 1 (0.5 × 7 mm²). (**b**) Specimen 2 (1.0 × 7 mm²). (**c**) Specimen 2 (2.0 × 7 mm²).

**Table 3.** Load case for leakage experiment.

| Specimen No. | Dimensions [mm] | Load Case | Fluid Weight Per Min [kg] |
|:---:|:---:|:---:|:---:|
| 1 | $0.5 \times 7$ | LC1 | 0.86 |
|   |   | LC2 | 1.62 |
|   |   | LC3 | 2.40 |
| 2 | $1.0 \times 7$ | LC1 | 3.51 |
|   |   | LC2 | 5.87 |
|   |   | LC3 | 7.02 |
| 3 | $1.0 \times 14$ | LC1 | 7.58 |
|   |   | LC2 | 12.49 |
|   |   | LC3 | 14.98 |

## 4. Numerical Simulation

The ANSYS software (CFX) was employed to solve steady state Navier–Stokes equations. A realizable k-ε turbulence model was selected using the software, both for its reliability with fully developed flows and for its capability to assure effective results in limited computational times. A Semi-Implicit Method for Pressure Linked-Equation (SIMPLE) algorithm was employed for pressure-velocity coupling, for a faster convergence, with second order accuracy. Analysis of the computational fluid dynamics model is based on a Eulerian domain model that employs the constant volume-based finite element method in the computational fluid dynamics (CFD) code [17,18].

### 4.1. Governing Equations

The fluid flow through the crack was used to simulate the flow of single-phase fluid in a pipeline and crack section. A CFD model was constructed using the volume of fluid (VOF) method for the leakage simulation. The VOF model was used in a RANS CFD analysis and included pressure boundary conditions on the fluid flow [18]. The governing equations for the fluid flow use the following mass and momentum conservation equations, expressed in Equations (2) and (3).

$$\frac{\partial \rho}{\partial t} + \frac{\partial}{\partial x_i}(\rho u_i) = 0 \tag{2}$$

$$\frac{\partial}{\partial t}(\rho u_i) + \frac{\partial}{\partial x_j}(\rho u_i u_j) = -\frac{\partial p}{\partial x_i} + \mu \frac{\partial^2 u_i}{\partial x_j \partial x_j} + b_i \tag{3}$$

where $b_i$ is the external body force; $p$ is pressure; and $u_i$ and $x_i$ are the Cartesian velocity and coordinate tensors, respectively. Only one momentum conservation equation is used for both fluids. The homogenous CFD approach uses the dynamic viscosity ($\mu$) and density ($\rho$) of the fluid mixture calculated using the volume fraction ($r_k$) with the constraint $\sum r_k = 1$, as expressed in Equations (4) and (5).

$$\rho = \sum \rho_k r_k \tag{4}$$

$$\mu = \sum \mu_k r_k \tag{5}$$

The standard k-e turbulence model was used in the CFD model to reflect the turbulence of the fluid flowing through the crack specimen.

### 4.2. Simulation Model

A node-centered finite volume method (CVFEM) based on the Eulerian–Eulerian model was used to simulate the behavior of the fluid flow in the leakage simulation. The fluid flow in the experimental conditions was modeled using a commercial Navier–Stokes VOF–CFD code in the CFX [13]. The CFD code converts the governing differential equation to a set of algebraic equations by discretizing the

fluid domain using CVFEM. The fluid leak simulation is based on the solution of the unsteady fluid flow assuming that the fluid flow is in steady state and mass conservation [14,15].

Leakage phenomena have been simulated considering the flow of fluid from the desired leakage location. The simulation model was prepared in consideration of the flow rate at the same crack sample location, reflecting the experimental conditions in Figure 4. In the leakage simulation, a steady-state leak rate is assumed as a part of the unsteady pipe flow simulation. Then, the initial steady-state head at the leak location is evaluated in the computational time step, and the leakage coefficient is evaluated. The procedure is carried out to compute the leak rate and the head at the leak location.

Before discussing the analogy between the orifice and the leakage, it would be useful to understand how the resulting equation for the leak rate computation can be used to simulate a leak along the pipeline. The leakage must be considered at a computational section. Due to the apparent similarity between a leak from a pipe through a crack and an orifice flow with respect to the static hydro pressure and internal pressure, the continuity equation at the leak location takes the following form expressed in Equation (6) [16].

$$Q_p - Q_{Leak} = 0 \tag{6}$$

where $Q_p$ is the flow rate in the pipe line, and $Q_{Leak}$ is the leakage flow rate.

The cross-sectional shape of the fluid flow with crack specimen is shown in Figure 9. The fluid domain of the CFD model was set around the cross section of the crack specimen. The geometry and boundary condition of the constructed CFD model are shown in Figure 10. The model was simplified and constructed with different crack areas for each specimen. The chosen mesh was composed of tetrahedral elements. To improve the accuracy of the calculation within the domain across the leak, a finer mesh was adopted in the neighborhood of the rectangular orifice as shown in Figure 11.

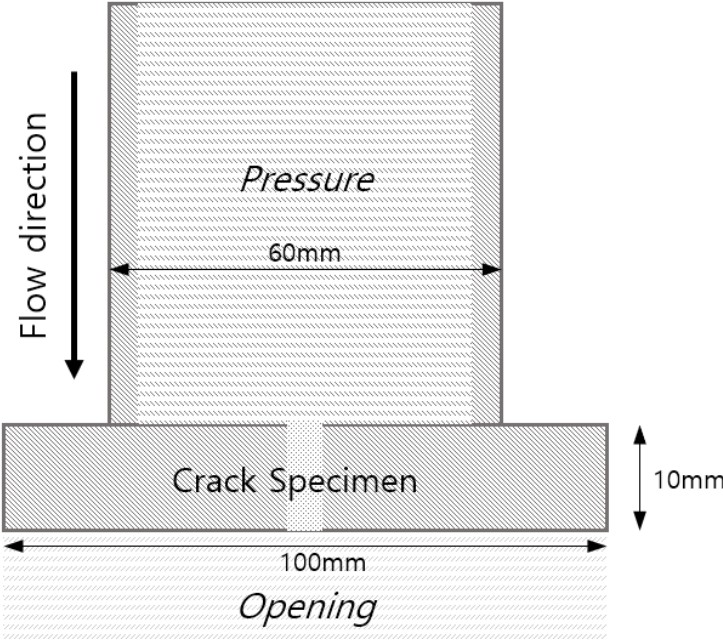

**Figure 9.** Geometry of leakage specimen section.

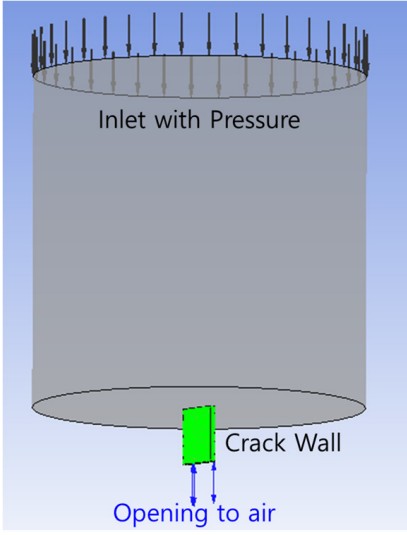

**Figure 10.** Boundary conditions of the CFD model.

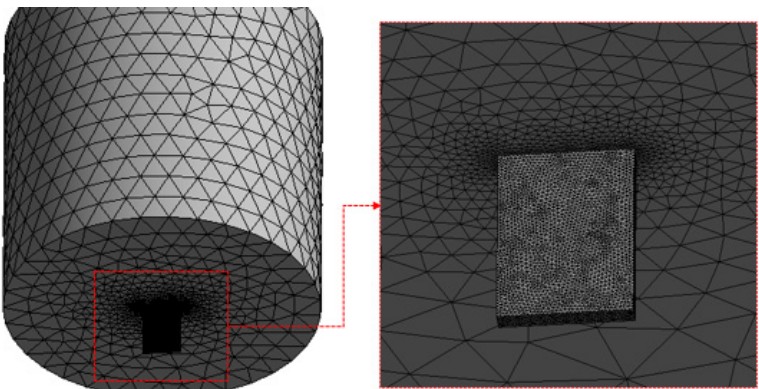

**Figure 11.** Mesh of CFD model.

To define the incompressible fluid flow in the CFD model, the boundary conditions are reflected in the fluid region of Figure 9, as shown in Figure 10. To reflect the pressure conditions inside the pressure tank from the experiment, the inflow part of the fluid was set to the inlet condition, and the pressure of each load condition was reflected. Furthermore, the region after the crack region was set so that the fluid passing through the crack specimen could be freely discharged by reflecting the opening condition to reflect the phenomenon of the fluid being discharged through the crack region. Additionally, it is assumed that the roughness factor of the crack surface is taken into account to prevent free slip from occurring, reflecting the rough wall conditions of the crack.

In the CFD model, only areas filled with water were modeled under experimental conditions to prepare the model considering the flow of single-phase flow fluids. To reflect the discharge of fluid into the air, the end surface of crack is assumed to be the similar condition as the experiment; the opening condition was applied considering the air effect and atmospheric pressure as shown in Figure 11. It shows the base condition of the CFD model. Figure 12 shows the grid model for CFD analysis. To properly simulate the flow of fluid in the crack, the size of grid was written by fine mesh. The number of grids used in the analytical model was approximately 20,000. The CFD model was written assuming a steady state condition for the analysis of leakage rate under each loading condition.

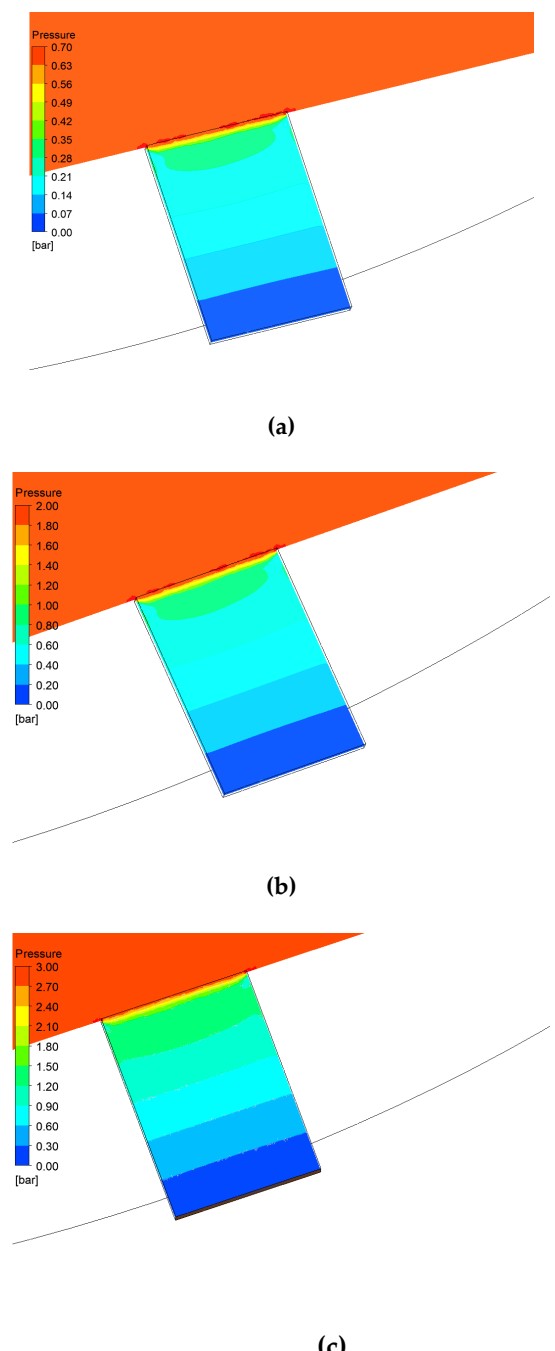

**Figure 12.** Pressure distribution for water flow of $0.5 \times 7$ mm$^2$ crack. (**a**) LC 1 (0.7 bar). (**b**) LC 2 (2 bar). (**c**) LC 3 (3 bar).

### 4.3. Simulation Model Evaluation Comparing with Experimental Results

The flow rate and the flow velocity distribution were calculated for each pressure condition and each crack specimen from the simulation analysis. For the results of this analysis, the material properties of the fluid applied for water are incompressible fluid, so density of water is 997 kg/m$^3$ and dynamic viscosity coefficient is $8.899 \times 10^{-4}$ kg/m-s [16]. The pressure distribution was calculated. To represent the CFD analysis results, plot the analysis results for Specimen No. 1 to confirm that the pressure drops rapidly as the fluid is released into the air while the pressure is distributed as shown in Figure 12. Additionally, the velocity distribution of fluids can be observed to see a significant increase in velocity at the area of leakage, as shown in Figure 13. Compare the results with the CFD results

for specimens 1, 2, and 3—the same conditions as the experiment; similar results were observed for all specimens. The leakage per unit hour was compared with the experimental results in Table 4 and Figure 14. The results of specimens 1, 2, and 3 were assessed to be similar, with some differences. The similarity of the results confirmed that the CFD model in this study was reasonable to calculate the amount of orifice leakage.

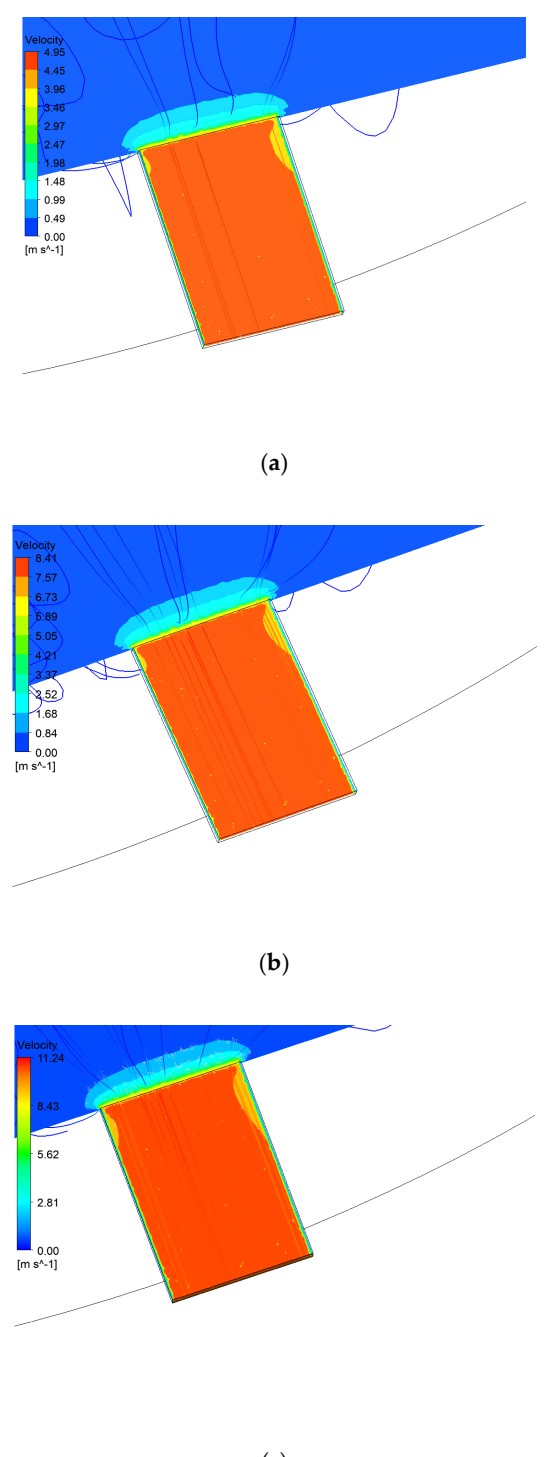

**Figure 13.** Velocity distribution and streamline for water flow of $0.5 \times 7$ mm$^2$ crack. (**a**) LC 1 (0.7 bar). (**b**) LC 2 (2 bar). (**c**) LC 3 (3 bar).

**Table 4.** Comparison between experimental and CFD results with water.

| Specimen No. | Dimension [mm] | Load Case | Water Leakage Rate [kg/min] | |
|---|---|---|---|---|
| | | | Experiment | CFD |
| 1 | 0.5 × 7 | LC1 | 0.86 | 1.15 |
| | | LC2 | 1.62 | 1.94 |
| | | LC3 | 2.40 | 2.37 |
| 2 | 1.0 × 7 | LC1 | 3.51 | 3.50 |
| | | LC2 | 5.87 | 5.93 |
| | | LC3 | 7.02 | 8.33 |
| 3 | 1.0 × 14 | LC1 | 7.58 | 8.08 |
| | | LC2 | 12.49 | 13.70 |
| | | LC3 | 14.98 | 16.90 |

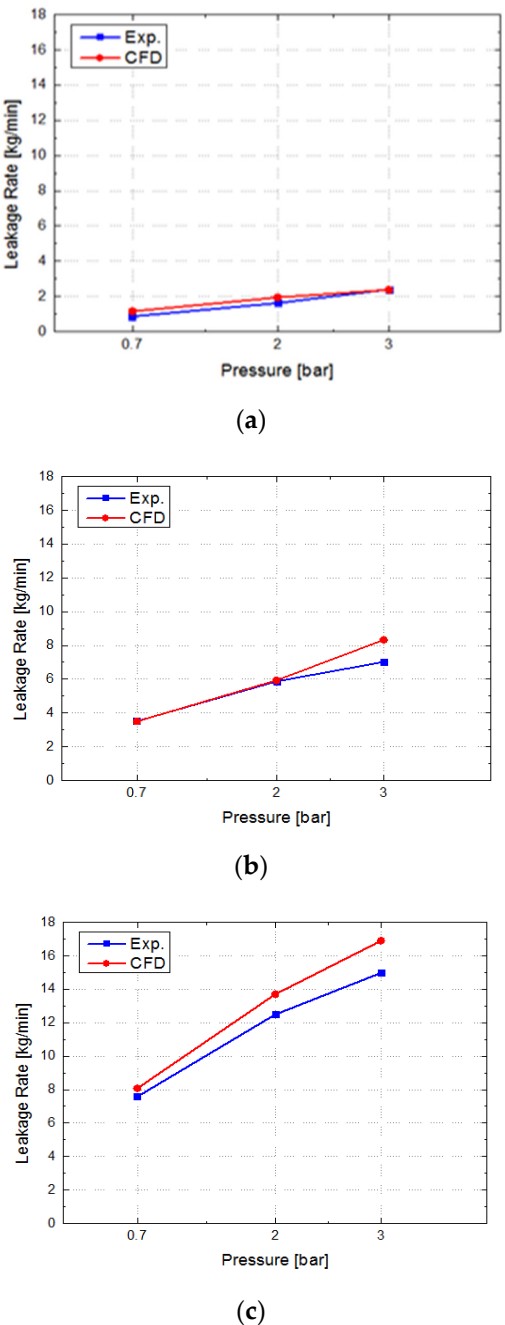

**Figure 14.** Comparisons of leakage rate between experiment and CFD result. (**a**) LC 1 (0.7 bar). (**b**) LC 2 (2 bar). (**c**) LC 3 (3 bar).

### 4.4. Simulation Result of LNG

Using the CFD model verified in the previous section, the leakage rate was calculated by applying the material properties of LNG (density: 498.5 kg/m$^3$ and dynamic viscosity coefficient: $4.45 \times 10^{-4}$ kg/m-s) [17]. The flow rate of LNG per minute was analyzed by the CFD model according to the crack area and loading condition. When the properties of LNG were applied, the leakage rate per minute was calculated for each condition, as shown in Table 5. The results show that they are similar to those applied to water. This is believed to be due to LNG's density and viscosity being lower than water under the flow conditions of incompressible fluid flow. The CFD analysis of LNG also represented the results of specimen No. 1. As shown in Figure 15, for each load condition, the pressure drops sharply as LNG is released into the air. The velocity distribution of fluids can also be observed to increase rapidly in the area of leakage as shown in Figure 16.

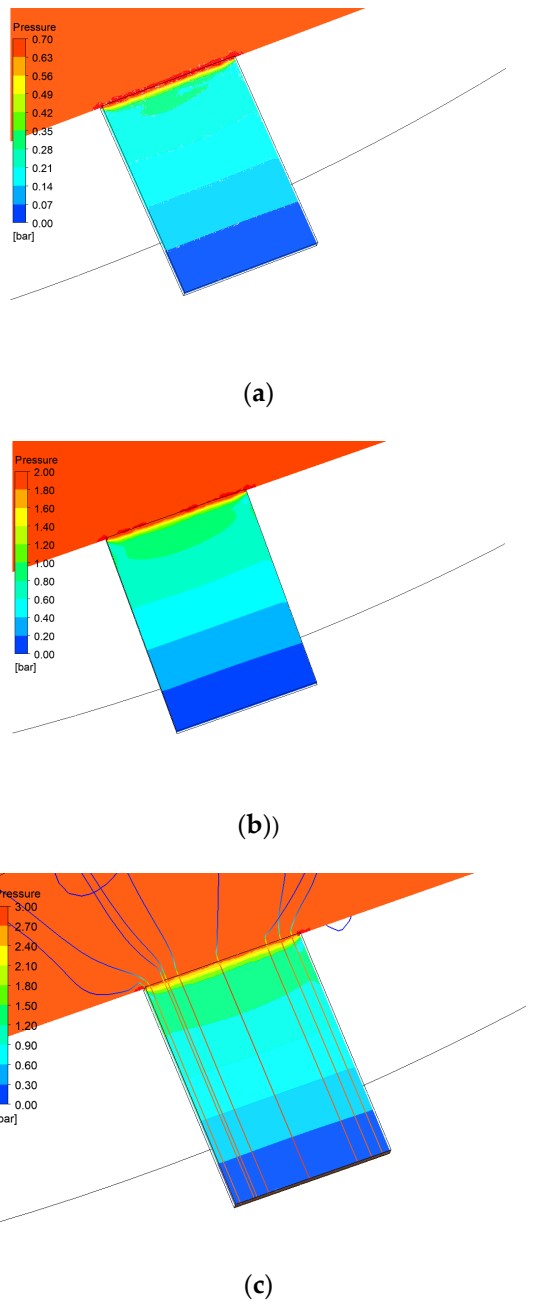

**Figure 15.** Pressure distribution for LNG flow of $0.5 \times 7$ mm$^2$ crack. (**a**) LC 1 (0.7 bar). (**b**) LC 2 (2 bar). (**c**) LC 3 (3 bar).

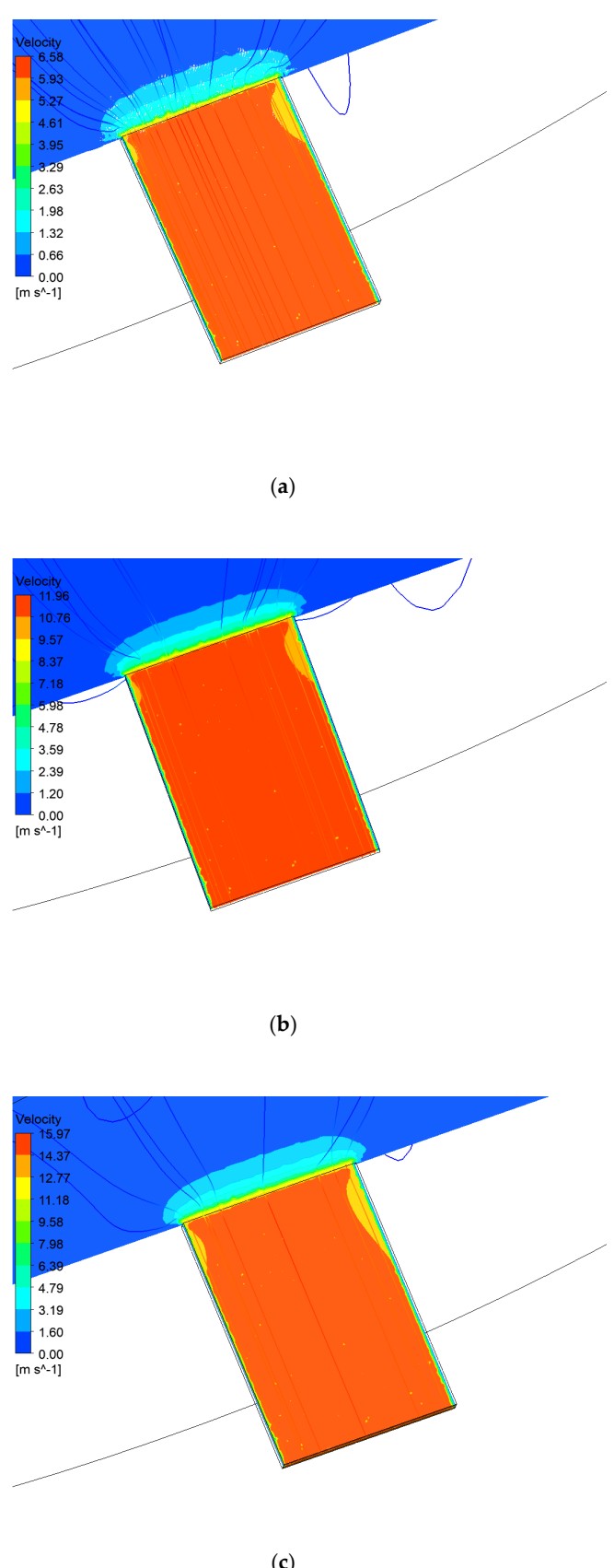

**Figure 16.** Velocity distribution and streamline for LNG flow of $0.5 \times 7$ mm$^2$ crack. (**a**) LC 1 (0.7 bar). (**b**) LC 2 (2 bar). (**c**) LC 3 (3 bar).

When the 3 pressure conditions were applied, the amount of LNG leakage was analyzed by the CFD method to the 5 case crack specimens as shown in Table 5. By analyzing the amount of leakage rate for each crack condition, as in Figure 17, it is confirmed that the amount of leakage increases nonlinearly as the area of the crack increases, and that the amount of leakage rate increases rapidly as the size of the crack increases. In addition, if the Reynolds number(Re) for the orifice flow is calculated in the 60 mm diameter pipe, which is the condition of the experiment and CFD analysis, and compared with the amount of leakage, as shown in Figure 18a, the amount of leakage would increase rapidly as the size of the crack increases. Depending on the aspect ratio (AR) of the orifice, it was confirmed that the amount of leakage flow increases in proportion to the area of the crack specimen, as shown in Figure 18b.

**Table 5.** Comparison between experimental and CFD results with water.

| Specimen No. | Dimension [mm] | Load Case | LNG Leakage Rate [kg/min] |
|---|---|---|---|
| 0 | 0.5 × 3.5 | LC1 | 0.32 |
|   |           | LC2 | 0.57 |
|   |           | LC3 | 0.73 |
| 1 | 0.5 × 7 | LC1 | 0.67 |
|   |         | LC2 | 1.17 |
|   |         | LC3 | 1.51 |
| 2 | 1.0 × 7 | LC1 | 1.40 |
|   |         | LC2 | 2.34 |
|   |         | LC3 | 3.11 |
| 3 | 1.0 × 14 | LC1 | 3.26 |
|   |          | LC2 | 4.98 |
|   |          | LC3 | 6.56 |
| 4 | 2.0 × 14 | LC1 | 9.33 |
|   |          | LC2 | 16.30 |
|   |          | LC3 | 20.38 |

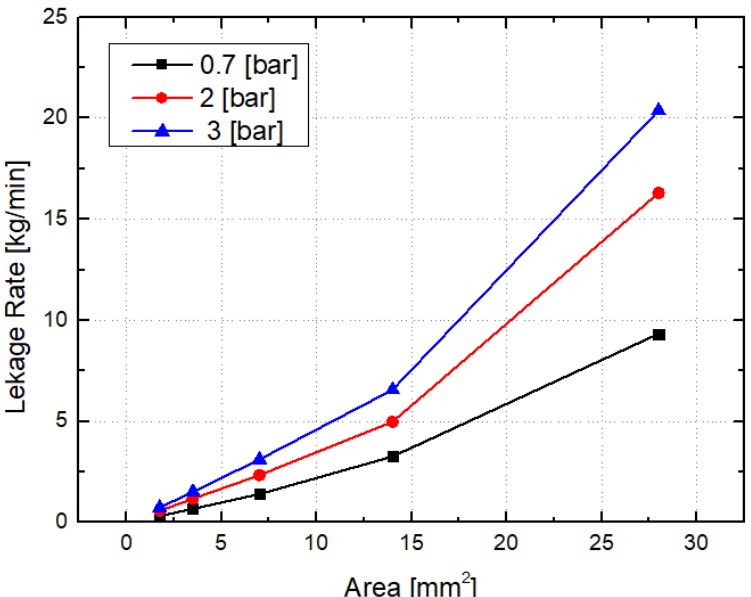

**Figure 17.** Orifice flow rate of each pressure conditions for rectangular crack.

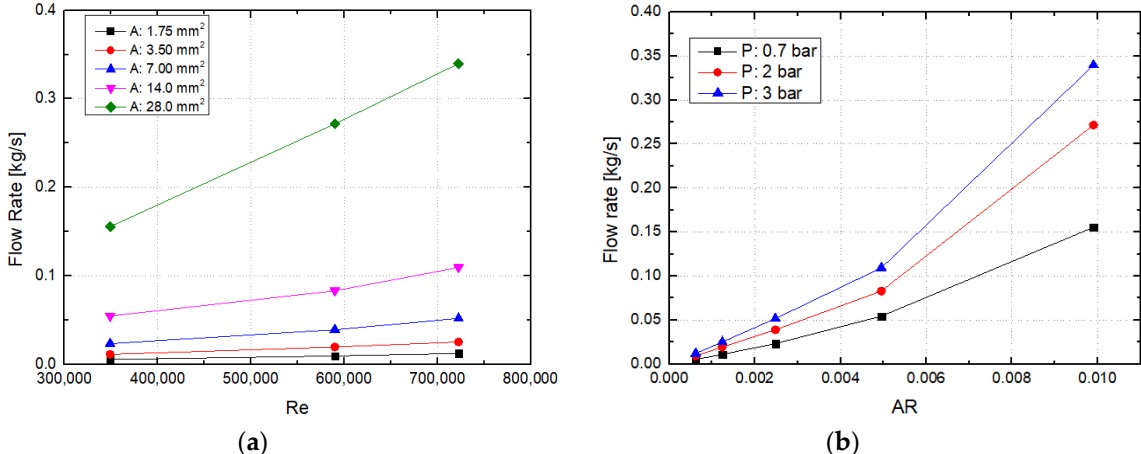

**Figure 18.** Orifice flow rate for Reynold number and aspect ratio. (**a**) Flow rate vs. Re. (**b**) Flow rate vs. AR.

## 5. Orifice Coefficient of LNG

The orifice coefficient of the LNG tank with the liquid cargo loaded was calculated while assuming the LNG loading conditions in Equation (1). The orifice coefficients of LNG are listed on Table 6. It is assumed that the density of the LNG is as 498.5 kg/m$^3$ and the LNG cargo with normal pressure head (25 m) is loaded in the tank. The results a listed in Table 5. Comparing the orifice coefficient under pressure conditions, as in Figure 17, it is found that the coefficient gradually increases as the internal pressure increases. When the area of the crack is less than 14 mm$^2$, it is possible to check that the orifice coefficient increases gradually due to the roughness of the wall of the crack wall. We can see that the area increased rapidly in sections over 14 mm$^2$. Based on this result, it can be confirmed that the orifice coefficient is distributed differently depending on the size of the crack as shown in Figures 19 and 20.

**Table 6.** Orifice coefficient according to crack area and pressure conditions.

| Specimen No. | Area [mm$^2$] | Load Case | Orifice Coefficient |
|---|---|---|---|
| 0 | 1.75 (0.5 × 3.5) | LC1 | 0.22 |
|  |  | LC2 | 0.30 |
|  |  | LC3 | 0.34 |
| 1 | 3.5 (0.5 × 7) | LC1 | 0.23 |
|  |  | LC2 | 0.31 |
|  |  | LC3 | 0.35 |
| 2 | 7 (1.0 × 7) | LC1 | 0.24 |
|  |  | LC2 | 0.31 |
|  |  | LC3 | 0.36 |
| 3 | 14 (1.0 × 14) | LC1 | 0.28 |
|  |  | LC2 | 0.33 |
|  |  | LC3 | 0.38 |
| 4 | 28 (2.0 × 14) | LC1 | 0.4 |
|  |  | LC2 | 0.54 |
|  |  | LC3 | 0.59 |

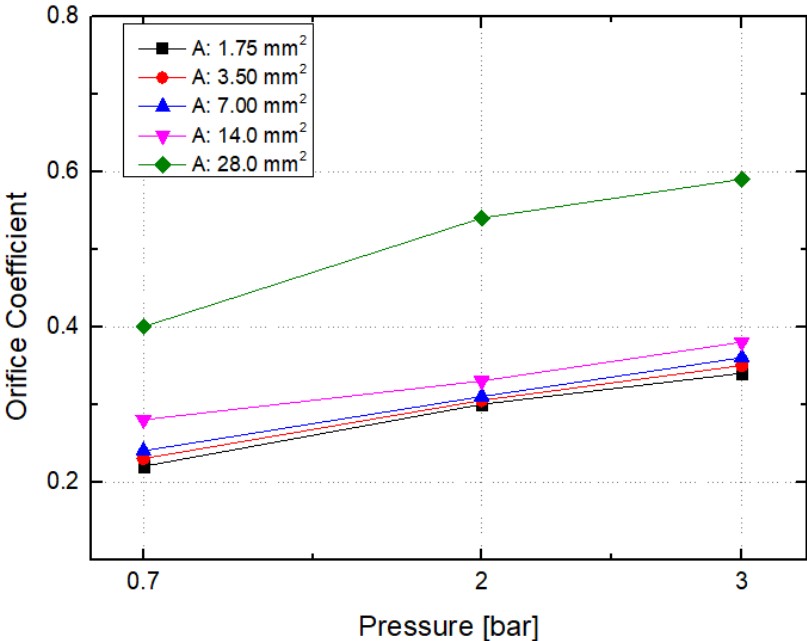

**Figure 19.** Orifice coefficients according to the pressure condition.

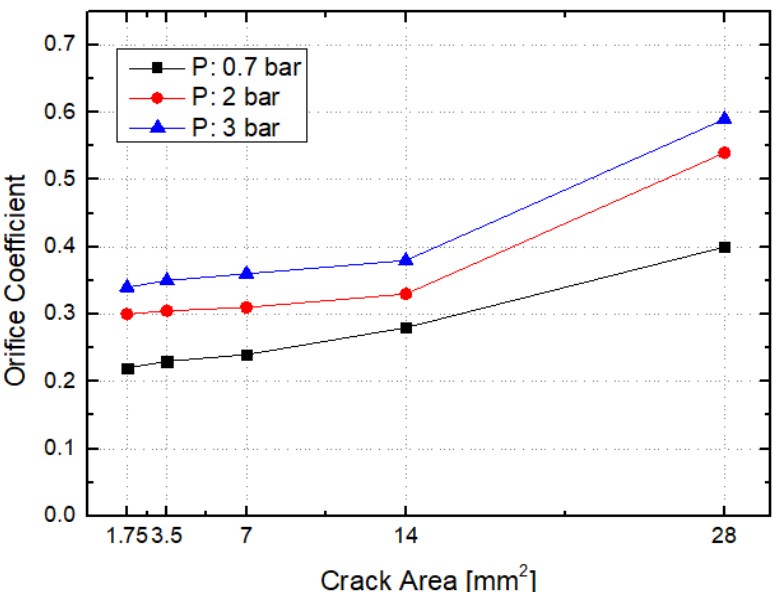

**Figure 20.** Orifice coefficients according to crack area.

## 6. Conclusions

The purpose of this study is to examine the validity of the design criteria applied in the design of liquid cargo storage systems of LNG vessels. This study examined the experimental and numerical results of the design of liquid cargo tanks, on the orifice coefficient of drip trays considered as a major design factor. In order to investigate the relationship between the leakage flow and the orifice coefficient in case of cracks in the tank, the distribution of leakage quantity was determined through an experiment in which the amount of leakage was directly measured for various crack areas by using a pressure tank filled with water. To ensure safety when designing the independent LNG tanks, a method of estimating the flow rate during drip tray design was studied. The flow rate of liquid cargo was measured by cracks using the pressure tank filled with water. Rectangular crack specimens were used, and changes in flow rate and orifice coefficients were investigated depending on the size of the crack.

Furthermore, to overcome an experimental limitation, the CFD method was applied to calculate the amount of leakage flow generated from the LNG filled tank by crack areas.

The orifice coefficient was found and calculated to vary with the cross-sectional area of the crack and the loading conditions of the liquid cargo. Therefore, the leakage from the cracks depended on the shape and size of the cracks. The orifice coefficients varied from 0.22 to 0.59 depending on the size of the cracks. Additionally, looking at the friction and viscosity effects of fluids at the point where the walls of the fluid and cracks were tangent, it was confirmed that the friction effects of the fluid were significantly issued below a certain crack size. In this study, it was found that the area of cracks was 14 mm$^2$ or less, which had a frictional and viscous effect. Furthermore, the effects were negligible in areas over 14 mm$^2$. This proves that if the area of a crack is less than a certain size, the amount of discharging fluid can vary depending on its area, size, and shape. It can also be explained that there is an effective size that viscosity and friction of the fluid affect, depending on the crack condition.

According to the guideline presented by the international guidance, it is described that a value of 0.1 for the orifice coefficient has been found to give good results compared with the test data [1]. However, the study confirmed that if the area of the crack is smaller than a certain size, the orifice coefficient changes nonlinearly due to the friction of the fluid. It was also found that the value of the coefficient was significantly greater than the value given in the guidelines. Based on the results of this study, it is, therefore, suggested that the orifice coefficient for the design of the LNG fluid tank needs to be reviewed.

**Author Contributions:** S.-Y.H. and J.-H.L. suggested the concept of orifice coefficient of fluid cargo tank of LNG vessels; K.-S.K. and H.-S.J. designed and performed experimental measurements; S.-Y.H. analyzed the data and suggested a CFD model for estimating the orifice coefficient; S.-Y.H. and J.-H.L. wrote the paper. All authors have read and agreed to the published version of the manuscript.

**Funding:** This research was funded by KIAT(Korea Institute for Advancement of Technology), No. 10063532.

**Acknowledgments:** This work was supported by the KIAT (Korea Institute for Advancement of Technology) grant funded by the Korea Government (Ministry of Trade Industry and Energy, No. P0001968; HRD program for R&D Human Resource Education Program for Green & Smart Ship and Industrial material core technology program, No. 10063532; development of steel application technologies against ice-induced crashworthiness and arctic temperature high toughness).

**Conflicts of Interest:** The authors declare no conflict of interest.

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
