# Peer review of "Experimental and Numerical Study of Orifice Coefficient of Cargo Tank Design of LNG Vessels"

_applsci, doi:10.3390/app10196667_

Round 1
Reviewer 1 Report
Application to LNG storage is the motivation for this work in investigating flow rate from cracks in storage tanks. Simple incompressible single phase approach is used. The simplifications this makes are not discussed. How representative is a rectangle to a typical crack?
Water flow is then considered. The experimental water leaks to air, wheras the CFD is single phase water, leaking to water. Again how significant is this? Would introducing thermo-fluid dynamics modify the behaviour significantly due to J-T effect with LNG?
The findings are presented in relation to the effect of crack area and pressure difference. Normal practice for determining flow coefficients would be to look at more meaningful parameters that would allow scaling between fluid media and various geometric configurations. For example, for incompressible flow the Reynolds number is the important factor (which explains the reduced C at small crack sizes). An appropriate reference length for scaling would be the hydraulic diameter of the rectangle. Thus other non-dimensional parameters would be rectangle aspect ratio, and length to hydraulic diameter ratio. Absolute area is taken into account as a linear factor in the coefficient equation. I think Re, AR and L/Dh are the three basic parameters varied in this study. It would be much better, and more scientifically rigorous to report in C as a function of these three parameters.
Limited configurations are presented for the 2 key independent geometrical parameters to be explored fully across a wide range. The sample manufacturing and test proceedure are not challenging and it would be easy to explore the relevant range more widely.
In its current form the work may be of interest to some readers, but it is very unlikely to be widely adopted/used. With the suggestions above it becomes more generic thus more widely applicable. With further work and presenting in a more scientific way (using the most appropriate reduced set of independent non-dimensional parameters) it could provide a useful data source on incompressible discharge coefficients of rectangluar orifices.
Author Response
I would like to thank you for your detailed and kind review so that the contents of the paper can be updated. The detail answer to your comment is attached as MS-WORD file below.
Thank you.

Reviewer 2 Report
The article analyzes the experimental and numerical results of the design of liquid cargo tanks for LNG, taking into account the orifice coefficient as the main design factor.
To investigate the relationship between the leakage flow and the orifice coefficient in case of cracks in the tank, the distribution of leakage quantity was determined based on an experiment in which the leakage size was measured directly for different areas of the crack with a pressure tank filled with water.
The fracture coefficient was calculated, changing depending on the cross-sectional area of the crack and the load conditions of the fluid charge.
In order to simulate the leakage of LNG, the leak rate was predicted by the Computational Fluid Dynamics (CFD) method using an orifice model.
In the absence of an appropriate model, the orifice model may be applied to provide a rough guide to the rate of release. The orifice model is presented in many references on consequence assessment, as well as most basic textbooks on fluid mechanics.
When using this model it is vital to keep in mind that the model does not reflect actual carrier construction very well, and the results should be interpreted as a rough guide to the rate of release for a given hole size.
Measurement with the orifice model is a model used very widely in the measurement technique, as it is a convenient and universal method, however, it has many limitations that should be taken into account if you want to correct the adopted orifice coefficient.
For example, the scope of application is limited for small flow rates and small cross sections due to the increasing error value. Another limitation is that this method can be used to measure steady flows or those whose flux changes slowly without sudden changes in flow.
As comments, I would include the lack of emphasis on the novelty in the article and the suggestion for further research on the verification of the orifice coefficient.
Literature review is not very long, many of item contain the same content.
These comments do not affect the content of the entire article and do not negate its validity.
Author Response
I would like to thank you for your detailed and kind review so that the contents of the paper can be updated despite the lack of content.
The answer to your kind comment is attached as a word file below.
Thank you.

Reviewer 3 Report
The authors use ANSYS FLUENT software for CFD simulation of the amount of leakage that occurs under pressure conditions of LNG tanks, but the boundary conditions are not presented very clear. So I suggest to use a table in which to have the experimental impute data in one side and in the other side the CFD boundary condition.
The results of the experimental study and of the CFD simulation are dependent of the strength of the material from which the tank is made?
Author Response

(The authors gave the same response as above.)
